# Molecular Alterations in Pancreatic Cancer: Transfer to the Clinic

**DOI:** 10.3390/ijms22042077

**Published:** 2021-02-19

**Authors:** Yolanda Rodríguez Gil, Paula Jiménez Sánchez, Raúl Muñoz Velasco, Ana García García, Víctor Javier Sánchez-Arévalo Lobo

**Affiliations:** 1Pathology Department, Hospital 12 de Octubre, Madrid, (Spain), Av. Córdoba, s/n, 28041 Madrid, Spain; yrodriguezg@salud.madrid.org; 2Molecular Oncology Group, Biosanitary Research Institute, Faculty of Experimental Sciences, Francisco de Vitoria University (UFV), Pozuelo de Alarcón, 28223 Madrid, Spain; paulajsanchez4@gmail.com (P.J.S.); raul.munoz@ufv.es (R.M.V.); ana.garcia@ufv.es (A.G.G.)

**Keywords:** molecular alterations, pancreatic ductal adenocarcinoma, molecular markers, therapy

## Abstract

Pancreatic ductal adenocarcinoma (PDA) is the most common cancer of the exocrine pancreas and probably the tumor that has benefited the least from clinical progress in the last three decades. A consensus has been reached regarding the histologic classification of the ductal preneoplastic lesions (pancreatic intra-epithelial neoplasia—PanIN) and the molecular alterations associated with them. Mutations in *KRAS* and inactivation of *CDKN2A*, *SMAD4* and *TP53* are among the most prevalent alterations. Next generation sequencing studies are providing a broad picture of the enormous heterogeneity in this tumor type, describing new mutations less prevalent. These studies have also allowed the characterization of different subtypes with prognostic value. However, all this knowledge has not been translated into a clinical progress. Effective preventive and early diagnostic strategies are essential to improve the survival rates. The main challenge is, indeed, to identify new effective drugs. Despite many years of research and its limited success, gemcitabine is still the first line treatment of PDA. New drug combinations and new concepts to improve drug delivery into the tumor, as well as the development of preclinical predictive assays, are being explored and provide optimism and prospects for better therapies.

## 1. Introduction

Pancreatic ductal adenocarcinoma (PDA) is a devastating disease with a survival rate of less than 10% [1]. This difficult prognosis is due to its late diagnosis—when the tumor has already metastasized—and its chemoresistance. The only curative therapy is surgery; however, this is only possible in 15–20% of cases. The impact of this disease is increasing, and it is predicted to become the second most common cause of death in the Western world [2,3]. It is therefore urgent to better understand the molecular bases of this disease in order to find better therapeutic options. The pathology of the pancreas is complex, and there are other pancreatic neoplasms such as mucinous papillary intraductal tumors, undifferentiated tumors, acinar cell carcinomas, cystadenomas and endocrine pancreatic tumors, but they are more infrequent and therefore we do not delve into them in this review.

Pancreatic cancer development shows a strong association with the consumption of tobacco in cigarettes, which increases the risk of contracting the disease by two or three times. This risk is proportional to the number of packs consumed annually. Other risk factors are diabetes, hereditary pancreatitis, chronic pancreatitis and exposure to ionizing radiation and chemicals, such as hydrochlorinated solvents used in some pesticides [4].

Based on morphological criteria, it is assumed that the cell origin of PDA is the ductal cell, which progresses from limited forms of neoplasia, such as pancreatic intra-epithelial neoplasia (PanIN), intraductal papillary mucinous neoplasia (IPMN) and cystic mucinous neoplasia (MCN), to an infiltrating carcinoma. However, experiments in mouse models have shown that the acinar cell can be a precursor for PDA development [5]. In this regard, inflammatory processes, such as pancreatitis, induce acinar cell dedifferentiation and subsequently trans-differentiation into a “ductal-like” cell, through a process known acinar to ductal metaplasia (ADM). This change is reversible, but when it is maintained, for example as a consequence of a chronic pancreatitis, it can lead to cellular transformation. The linear model that explains the origin and progression of PDA is similar to the one described for the colon, where a series of molecular alterations correlate with the degrees of dysplasia associated with precursor lesions (PanINs). PanIN-1A and 1B are characterized by the presence of elongated of ductal cells and mucus production (mucinous metaplasia), mild nuclear atypia and *KRAS* mutations. PanIN-1B adds to the mucinous metaplasia the formation of papillae or micro papillae; it keeps the atypia slight and molecularly shows inactivation of *CDKN2A*. PanIN-2 shows a higher degree of dysplasia and stack nuclei that look hyperchromatic and lose polarity. The presence of atypical mitosis towards the luminal pole is related to greater dysplasia and molecularly correlates with *CDKN2A* loss of function. Finally, PanIN-3 is characterized by *TP53* and *SMAD4* inactivation and has been considered a carcinoma *in situ*. However, recent evidence relates this lesion to tumor infiltration, indicating a possible mischaracterization. More extensive studies are required to better characterize this stage using PanIN-3 at less or more advanced stages. This linear model is currently under debate, and Notta et al. demonstrate that genomic instability due to chromotripsis has an important role in PDA genetics specifically at early stages [6].

Intraductal papillary mucinous neoplasm (IPMN) maintains the majority of alterations associated with PDA except *SMAD4* losses and differentially presents mutations in *PIK3CA*, *GNAS* and *RNF43* [7]. The other precursor lesions involved in PDA development are mucinous cystic neoplasm (MCN) and intraductal tubular papillary neoplasm (ITPN) (Table 1). MCN is molecularly characterized by a lower degree of loss of heterozygosity (LOH) in relation to PanINs and a lower number of mutations than IPMN, which could be related to a better prognosis [8]. Finally, the less frequent precursor ITPN, which is associated with higher risk of PDA development, very rarely presents mutations in *KRAS*, *NRAS* and *GNAS* but shows *PIK3CA* mutations and alterations of the AKT and mTOR pathway [9].

New studies of next generation sequencing (NGS) can be used to describe new molecular alterations and update PDA classification in different subgroups with different prognosis and responses to treatment. This better classification will undoubtedly help the best management of this type of tumor, adjusting it to the needs of each patient.

## 2. PDA Genetic Alterations

Tumor progression is the consequence of the accumulation of multiple genetic alterations that disrupt normal biological pathways. Oncogenic mutations, due to point mutations, amplifications or translocations, are responsible of the different tumor hallmarks, such as sustained proliferation, metabolic reprogramming, angiogenesis, inflammation and metastasis. Loss of function in tumor suppressor genes due to mutations, deletions or promoter inactivation contributes to tumor progression through the alteration of pathways involved in DNA repair and apoptosis that leads ultimately to genome instability.

Oncogenes and tumor suppressors are equally altered in PDA, and recent NGS studies have shown the enormous complexity in the number of molecular alterations present in the PDA that range from somatic mutations (*KRAS*, *TP53*, *ARID1A*, *TGFBR2*, *RREB1* and *PBRM1*), germline alterations (*BRCA1*, *BRCA2*, *ATM* and *PALB2*) and changes in copy number with gains (*c-MYC*, *ERRB2* and *KRAS*) and losses (*CDKN2A*, *SMAD4*, *ARID1A* and *PTEN*). Of all these alterations, those that occur in *KRAS*, *CDKN2A*, *TP53* and *SMAD4* stand out as the most prevalent, either through mutations or variations in copy number. In addition, a large number of alterations with lower prevalence are responsible for the heterogeneity of this tumor type. In this group, we highlight genes involved in DNA repair, cell cycle regulation, the TGF-*β* pathway, chromatin regulators and axonal guidance [10,11,12,13]. It is necessary to highlight the presence of an important desmoplastic reaction in this type of tumor—up to 90% of the tumor volume can be stroma—which have made the task of identifying genetic alterations particularly difficult [14].

### 2.1. KRAS

Activating mutations in *KRAS*, usually in codon 12 (G12D, G12V and G12R), is considered the driver mutation, and it is the first recurrently mutated gene present in nearly all PDA [15]. Besides point mutations, there are amplifications in the 12p12.1 region, where the gene is located [7,13]. Chan-Seng-Yue et al. demonstrate how *KRAS* mutant dosage defines different pancreatic cancer phenotypes; higher dosages are related to a more undifferentiated and aggressive phenotype than lower, which progress differently through the acquisition of other oncogenic gains such as *MYC* amplifications [16,17]. *KRAS* mutants activate PI3K and MEK signal transduction pathways and the transcription factors c-JUN and c-MYC, both potent inducers of cell proliferation, but also support pancreatic growth through the regulation of nucleotide synthesis [18].

Conditional mouse models using *Pdx1-Cre* or *Ptf1a-Cre* have shown the importance of *KRas* mutations, mainly G12V and G12D, as an initiating event in PDA and have unraveled the role of both acinar and ductal cells in PDA development. In these models, KRas activation in acinar cells induces a high frequency of low-grade mouse PanINs (mPanINs) compared with ductal cells that later evolve into high-grade mPanINs. On the contrary, ductal cells are quite refractory to *KRas* mutant and lead to no mPanINs or very few. To be fully transformed, acinar cells expressing mutant *KRas* require heterozygous mutation in *Tp53*, while ductal cells require homozygous mutations. When transformation occurs, mPanINs derived from ductal cells give rise to more aggressive tumors compared with acinar derived, supporting that acinar cell tumorigenesis is associated with low-grade mPanINs [19]. KRas^G12D^ expression in acinar cells concomitantly with the induction of pancreatitis increased the frequency of PDA formation [5]. Inflammatory processes can induce acinar cell dedifferentiation and its subsequent trans-differentiation through ADM, which favors cell transformation. In this progenitor state, KRas activates inflammatory pathways to initiate pancreatic cancer [20]. KRas drives the expression IL17 receptor and type I cytokine receptor complexes (IL2r*γ*-IL4r*α* and IL2r*γ*-IL3r*α*1) to establish a “hematopoietic-to-epithelial signaling axis” and enhance mPanIN progression and metabolic reprogramming [21,22].

### 2.2. CDKN2A

Experiments in mouse models clearly demonstrated the suppressive role of the *CDKN2A* locus in the development and progression of PDA [23]. This locus encodes two tumor suppressors (p16^INK4^ and p14^ARF^) through different initial exons and reading frames and with different biological functions. While p16^INK4^ is a CDK4/CDK6 inhibitor, p14^ARF^ sequesters MDM2, which targets p53 for degradation. Their loss is usually observed in moderately advanced lesions with some dysplasia (PanIN-1B, 2 and 3). *CDKN2A* loss of function occurs in 70–80% of cases and may result from mutations and/or losses of the wildtype allele (40%), homozygous deletions (40%) [24] or promoter hypermethylation (20%) [13,25]. Due to this physical juxtaposition and the frequent homozygous deletion of the locus, many pancreatic tumors lose both suppressors, which leads to the inactivation of the retinoblastoma (Rb) and p53 pathways. However, mutations only affect p16^INK4^, suggesting its prominent role in PDA.

### 2.3. TP53

*TP53* mutations in the DNA binding domain occur in approximately 50–70% of PDA cases and with variable frequency in PanIN-3 lesions, supporting the idea that PanIN-3 can show different levels of malignancy and PDA subtypes [16,26,27]. The loss of p53 function constitutes a double threat, since it results in the loss of cell cycle control and in the deregulation of programmed cell death, leading to the survival and proliferation of cells with chromosomal alterations. PDA tumors present a high frequency of copy number aberrations, aneuploidy and complex chromosomal rearrangements as a consequence of genomic instability and genome duplication during tumor progression [12,16]. Chan-Seng-Yue et al. recently described that *TP53* losses are more prevalent in specific molecular subtypes (basal-like; see molecular classification) correlating with a higher metastatic potential and poor response to chemotherapy [16]. *TP53* losses activate the JAK2/STAT3 signaling pathway to promote pancreatic tumor growth and resistance to gemcitabine treatments, which correlates with poor prognosis and reduced patient survival [28].

Besides the already known functions of p53 as a guardian of the genome, recent evidence has demonstrated its implication in other biological processes such as splicing. PDA cells harboring *TP53* missense mutations exhibited aberrant use of exons compared with wild-type or harboring truncating mutations. In particular, the TP53^R175H^ mutant regulates the expression of specific GTPase-activating protein isoforms (GAPs) in pancreatic cancer as a consequence of altered alternative splicing [29]. Other hotspot mutants, such as TP53^R273H^, which inhibits the expression of *KLF6*, promote migration and metastasis [30]. Moreover, TP53^R273H^ and TP53^R175H^ mutants can promote metabolic reprogramming in pancreatic cancer by preventing the nuclear translocation of GAPDH and enhancing glycolysis [31]. The KPC mouse model supports the importance of p53 during tumor progression, and mice expressing TP53^R172H^ concomitantly with KRas^G12D^ in the pancreas (*Pdx1-Cre*) develop highly metastatic pancreatic tumors through the upregulation of PDGFRb, mimicking the human disease [32,33].

### 2.4. SMAD4

The *SMAD4* gene encodes a transcriptional regulator that constitutes a central element in the transforming growth TGF-*β* (TGF-*β*) signaling pathway. It is subjected to homozygous deletions (50%) and inactivating mutations (10–20%) [34]. *SMAD4* inactivation is a late event, detected only in PanIN-3 lesions and invasive tumors, so it is considered to be linked with tumor progression and worse prognosis [27,35]. *SMAD4* loss affects the interaction with the microenvironment rather than cancer cell growth, and its restoration in pancreatic cancer cell lines has a minimal impact on proliferation in vitro but impairs its ability to form tumors in immunocompromised mice because of less angiogenesis and remodeling of the extracellular matrix [36]. Mutations in *SMAD4* are associated with metastases and correlate with a worse prognosis [37]. *SMAD4* loss renders pancreatic cancer resistance to radiotherapy due to ROS induction and promotion of autophagy [38]. To favor metastasis, *SMAD4* loss induces epithelial–mesenchymal transition (EMT) and metabolic reprogramming through the translocation of phosphoglycerate kinase 1 (PGK1) to the nucleus, where it is able to repress E-cadherin and favor EMT [39].

Mouse models have demonstrated that SMAD4 is dispensable for normal pancreas development but is critical for pancreatic cancer progression [40]. Its inactivation accelerates KRas^G12D^ dependent pancreatic cancer [41].

### 2.5. c-MYC

The *c-MYC* oncogene is a transcription factor and a potent driver of cell proliferation overexpressed in a large number of tumors [42]. PanIN-2, PanIN-3 and PDA overexpress c-MYC, independently of the mutational load. Its expression is associated with the squamous molecular subtype and a worse prognosis [7] (Table 2). The increase in c-MYC levels may be due to KRAS activation, *SMAD4* loss and/or amplifications of the 8q24.13 locus [13]. Bhattacharyyra et al. recently described how the secretion of FGF1 by cancer associated fibroblasts (CAFs), present in the tumor microenvironment, can sustain high c-MYC levels in pancreatic tumor cells [43].

Mouse models have been extremely useful in deciphering the role of c-MYC in pancreas homeostasis and PDA development. c-MYC is required for pancreas development, and its downregulation is required for complete acinar differentiation [44,45]. Its overexpression in the acinar compartment (*Ela1-Myc* mouse) induces ADM and transformation [45]. Its co-expression concomitantly with Kras^G12D^ represses the type I interferon pathway and enhances the expression of cytokines and chemokines that generate an immune suppressive microenvironment [46,47,48]. Importantly, some of these results were obtained with endogenous c-MYC levels, suggesting that c-MYC can participate in tumor progression without being overexpressed and supporting its importance in pancreatic cancer biology.

### 2.6. GATA6

GATA6 is a pioneer transcription factor that regulates the maintenance of the acinar identity [49]. It is frequently altered in PDA due to overexpression and copy number alterations with an impact on prognosis [50]. This role has been validated in mouse models, where KRas^G12D^ and Gata6 cooperate to drive pancreatic tumorigenesis [51].

Grainne et al. observed that GATA6 expression is associated with specific molecular subtypes with prognostic implications [52]. Gains in copy number correlate with a better outcome, while low GATA6 levels due to chromosomal loss or promoter hypermethylation correlate with a worse evolution (Table 2) [10,13,16,51,53]. Mechanistically, GATA6 is necessary for the maintenance of a differentiated state and the inhibition of epithelial–mesenchymal transition (EMT). Its loss activates the EMT program, promoting tumor progression and resistance to adjuvant chemotherapy and radiotherapy with lower overall survival [54].

## 3. PDA Epigenetic Alterations

Gene expression is regulated by a complex set of modifications—acetylation, methylation and phosphorylation, among others—that affect histones, on which the DNA is wrapped forming the nucleosomes. A specific group of proteins named chromatin modifiers and chromatin remodelers introduce, erase or read these modifications to regulate gene expression. Chromatin remodelers (readers) displace nucleosomes in order to allow or repress transcription.

DNA methylation is the best characterized modification described for DNA at the moment. DNA methylation patterns, especially in CpG islands, are associated with gene expression silencing. In a recent study published by The Cancer Genome Atlas (TCGA) on 150 cases of PDA, the integration of DNA methylation and mRNA expression showed 98 genes silenced by methylation, some of them previously involved in other tumor types but not described in PDA, such as *ZFP82*, *PARP6* and *DNALC15*, which were identified in breast cancer as chemo-resistance associated genes [13]. Despite these groups of genes, there is no evidence of a specific subtype of PDA with hypomethylation.

Mutations in gene coding for histone modifiers—*MLL2*, *MLL3*, *SETD2* and *KDM6A*—are frequently mutated in 24% of PDAs. *KDM6A* mutations have been described mainly in squamous pancreatic tumors, correlating with worse prognosis [10,12]. In mouse models, *KDM6A* loss induces squamous-like metastatic pancreatic cancer through aberrant activation of super-enhancers that regulate Δ*Np63*, *c-MYC* and *RUNX* [55].

Histone marks allow the recruitment of chromatin remodelers, the function of which is the displacement of nucleosomes to regulate transcription; 14% of PDAs have mutations in different subunits of the SWI/SNF chromatin remodeling complex, highlighting those that occur in *SMARCA4* (*BRG1*), *SMARCA2* (*BRM*) and *ARID1A* [10,13]. Mouse models have shown the complexity of its function and its dependency on the epigenetic context to work as a tumor suppressor or oncogene. In the conditional mouse model *Pdf1a-Cre*, *Brg1* loss in acinar cells cooperates with Kras^G12D^ at early stages to form cystic neoplastic lesions similar to intraductal papillary mucinous neoplasia (IPMN), which progresses to a less aggressive form of PDA without mPanINs [56] due to the role of Brg1 in “acinar to ductal dedifferentiation” [57]. In duct cells, Kras^G12D^ activation and *Brg1* deletion induce a “dedifferentiated ductal state” that favors IPMN. However, its re-expression during later stages of neoplasia (IPMN) favors PDA development [58]. Thus, *Brg1* can act as a tumor suppressor or oncogene depending on the epigenetic context.

ARID1A is another subunit of the SWI/SNF complex, which appears mutated or deleted in 12–23% of PDAs [7]. In acinar cells, *ARID1A* loss is necessary for Kras^G12D^ induced transformation [59], and its absence is associated with acinar homeostasis alteration in response to pancreatitis, poor cell differentiation and tumor formation with a more aggressive mesenchymal phenotype [60,61]. In duct cells, *ARID1A* loss correlates with ductal dedifferentiation and in the presence of Kras^G12D^ facilitates the development of intraductal papillary mucinous neoplasia (IPMN) and PDA [62].

The characterization of all these alterations has allowed us to explore new vulnerabilities of PDA tumors, a strategy known as “synthetic lethality”. Tumors with *ARID1A* deficiency show a greater sensitivity to ARID1B inhibition, and tumors deficient in *SMARCA4* have a high vulnerability to SMARCA2 inhibition [63,64]. ARID1A is necessary in DNA damage response and is recruited actively to the damaged region. ARID1A-deficient tumors have deficiencies in DNA repair processes and are more sensitive to poly ADP ribose polymerase (PARP) inhibitors [65]. *KDM6A* loss sensitizes pancreatic tumors to bromodomain and extra terminal (BET) domain inhibitors and histone deacetylase (HDAC) inhibitors [55,66].

## 4. PDA Molecular Subtypes

Massive sequencing studies have shown the heterogeneity and complexity of PDA molecular alterations beside the classical four: *KRAS*, *CDKN2A*, *SMAD4* and *TP53*. These include less prevalent mutations, changes in copy number and chromosomic alterations as a consequence of genome instability and genome doubling. The first attempt to obtain a molecular characterization was carried out by exon sequencing in 24 PDA cases, wherein 65 mutations were described and grouped into 12 molecular pathways. This analysis was later extended by Biankin et al. over 100 PDA cases but was insufficient to establish molecular subtypes [11,67]. Collisson et al. described the first molecular classification in 27 PDA cases using a microarray analysis to define three subtypes: (i) classical, characterized by the expression of epithelial genes such as *GATA6*; (ii) quasimesenchymal, characterized by the expression of mesenchymal genes and worse prognosis and (iii) exocrine, characterized by the expression of acinar genes [53]. A different strategy was used by Moffitt et al., who carried out a transcriptomic study based on a virtual dissection to distinguish molecular alterations present in the epithelium or in the stroma. With this approach they described two major groups that simplified Collisson’s classification: (i) classical and (ii) basal [68]. Chan-Seng-Yue et al. using laser capture microdissection (LCM), whole-genome sequencing and whole-transcriptome sequencing split those classifications in two, classical A and B and basal-like A and B, and showed the importance of genome duplications and *KRAS* dosage in the different subtypes [16]. Other studies have emphasized the genomic alterations present in this type of tumor. Waddell et al. combined massive sequencing and chromosomal rearrangements and proposed four subtypes: (i) stable, characterized by the presence of aneuploidy and few structural chromosomal variations; (ii) with local rearrangements, with focal alterations and copy number changes; (iii) with dispersed rearrangement, with a moderate range of chromosomal alterations; (iv) unstable, with a high number of structural variations in addition to mutations in *BRCA1*, *BRCA2* and *PALB2*—genes involved in DNA repair [12]. Bailey et al. conducted the study with the highest number of cases so far; 456 PDA cases were analyzed using massive sequencing, deep exome sequencing and copy number analysis. This analysis identified four subtypes: (i) squamous with high frequency of *TP53* and *KDM6A* mutations in addition to *TP63*Δ*N* overexpression and *GATA6* promoter hypermethylation; (ii) progenitor, with a high expression of transcription factors involved in pancreatic development (FOXA2/3, PDX1); (iii) immunogenic, characterized by significant inflammation with infiltrates of B and T cells; (iv) aberrantly differentiated endocrine/exocrine (ADEX), characterized by the overexpression of genes related to pancreatic lineage differentiation [10].

A huge handicap of these studies, except the work carried out by Moffitt et al., was the low tumor cellularity due to the high degree of stromal cell infiltration. In this regard, the latest study published by The Cancer Genome Atlas (TCGA) showed this effect in the different classifications and in their overlapping (Table 2). The squamous and basal subtypes overlap as well as the progenitor and classical, but this study suggests that the exocrine, immunogenic and ADEX subtypes are a consequence of the low tumor cellularity and might not be real [13] (Table 2). These results were validated by Puleo et al. in a series of 309 PDA cases that were analyzed by DNA sequencing and transcriptomic profile. By deconvoluting normal, tumoral and microenvironment transcriptomic signals present in the tumor microenvironment, the authors identified five different subtypes with prognostic value: (i) pure basal-like; (ii) stroma activated; (iii) desmoplastic; (iv) pure classical and (v) immune classical. This work also supports the idea that the ADEX subtype is a possible contamination by the normal exocrine pancreas rather than a real subtype [69].

We hope that a better characterization of the different molecular subtypes will be transferred to the clinic to define more specific treatments in the future.

## 5. Intratumoral Heterogeneity and Metastasis

Pancreatic cancer is an aggressive disease; the main cause of death is the presence of metastases, detected in 52% of cases at the time of diagnosis. It has been estimated that the transformation process lasts about ten years until the establishment of a non-metastatic primary tumor and five years for the generation of metastases. From this moment, the life expectancy is about two years [70]. However, the dissemination process might begin at early stages, Rhim et al. have demonstrated using the KPC mouse models the presence of PDA cells in the bloodstream followed by liver colonization even before the emergence of the bulk tumor [71]. This suggests that different genomic alterations allow the early appearance of clones with metastatic potential.

There are few studies that correlate molecular alterations in primary tumors with their corresponding metastases. Frequently, at the time of diagnosis the tumor has already metastasized, and patient’s poor condition does not allow surgery. Next generation sequencing studies have demonstrated the existence of different parental clones in the primary tumor, which develop as a result of the acquisition of new genetic alterations, with different metastatic potential [70]. Although limited mutational heterogeneity has been observed between primary and metastatic tumors [72,73], which might suggest the importance of other non-genetic mechanisms, we must consider the technical limitations of those studies based on primary tumors with low cellularity. Chan-Seng-Yue et al. solved this problem using laser capture microdissection to perform whole-genome sequencing and whole-transcriptome sequencing to demonstrate the presence of intratumoral heterogeneity and the coexistence of different molecular subtypes in the primary tumors—classical and basal. The basal subtype is associated with the stage and metastatic potential, but both subtypes harbor *KRAS* imbalance in 71% of their metastasis as a consequence of genome doubling, indicating that *KRAS* dosage and polyploidization are driving forces of metastasis [16].

Campbell et al. analyzed gene rearrangements in primary pancreatic tumors and their corresponding metastases, concluding that there is a pattern of specific rearrangements for PDA, with higher prevalence of deletions and inversions and less of duplication and amplicon rearrangement. Some of these rearrangements are already present in the primary tumor, and others are subsequently acquired, correlating with the colonization of specific organs. For example, *KRAS* amplification occurs mainly in peritoneal metastases, while *PARK2* losses and *c-MYC* amplifications arise in pulmonary metastases [74]. *SMAD4* loss have been correlated with tumor spread and worse prognosis, regardless of tumor size, grade and lymph node involvement [35,75], but we must consider that its loss is associated with the classical subtype; most basal subtypes retain *SMAD4* expression, suggesting that alteration in other pathways may lead to the same outcome [16].

In addition to these genetic alterations, there are epigenetic modifications that arise as a result of clonal evolution within the primary tumor. These alterations include changes in histone methylation and acetylation patterns and changes in DNA methylation in heterochromatic and euchromatic regions that correlate with a metabolic reprogramming towards oxidative pathways [76]. Connecting metabolism with invasion, Anderson et al., in a review of 143 PDA tumors, including metastases, found overexpression of Hexokinase 2 (*HK2*) and its correlation with a worse prognosis and lower survival. In cell lines, overexpression of HK2 is correlated with increased proliferation, invasion and metastasis [77].

## 6. Genetic Alterations Associated with Familial Pancreatic Cancer

Approximately 5–10% of patients diagnosed with pancreatic cancer have a genetic basis and are considered familial pancreatic cancer (FPC) patients. In this group, we include families with two or more first-degree family members affected by PDA. Some of them correspond to known syndromes with germline mutations in genes associated with cancer predisposition syndromes, such as *BRCA2* [78], *PALB2* [79] and *ATM* [80] (Table 3). In other cases, they correspond to family groups without knowing the inherited mutation. The presence of one member affected increases the risk of PDA by 2 or 3 times, 6 times if there are two family members and 32 times if there are three [81]. The inheritance pattern found in these cases is autosomal dominant with a variable penetrance. However, in most cases of FPC the genetic cause is still unknown.

Genome-wide association studies (GWAS) have detected associations between Single nucleotide polymorphisms (SNPs) and specific phenotypes. SNPs in the *TERT* gene, the orphan nuclear receptor *NR5A2* and others present in regions 13q22.1 and 15q14 (without association with known genes) have been associated with a higher risk of PDA [82]. NR5A2 is an orphan nuclear receptor involved in a large number of biological processes. In the pancreas, NR5A2 has different expression patterns, contributing first to pancreas development and later maintaining the acinar phenotype. Nr5a2 contributes to the complete acinar differentiation through the direct regulation of the nuclear liver factor alpha 1 (Hnf1a) [83]. Analysis of polymorphism in the proximity of the *NR5A2* gene correlates with a reduction in protein levels and the development of PDA. The stratification of patients based on *NR5A2* expression allows the distinction in two groups, with an association between low levels of NR5A2 and a higher prevalence of chronic pancreatitis and PDA development [82,84]. Finally, alterations in the locus 9q34 (SNP rs505922), which codes for the first intron of the ABO blood group, is associated with an increased risk of pancreatic cancer and correlates with the epidemiological findings of higher incidence of PDA in groups A and B than in 0 [85].

## 7. Molecular Alterations as a Predictive Response Factor and New Targets

PDA treatment should be multidisciplinary, with the aim of attaining free margins after tumor resection. According to these criteria, PDAs can be: resectable; “borderline” resectable; locally advanced unresectable and unresectable with metastases [86]. Currently, in the case of resectable tumors, the best therapeutic option is surgery. Unfortunately, only 15–20% of cases are potentially resectable or borderline at the time of diagnosis. Regardless of the quality of the surgery, up to 70% of tumors, initially classified as resectable present surgical margins and a high rate of recurrence, locally and at distant sites. Therefore, the use of adjuvant chemotherapy for all resected pancreatic tumors without prior neoadjuvant therapy (including T1N0) is considered, using gemcitabine-based protocols (DNA synthesis inhibitor), gemcitabine with capecitabine or 5-Fluoruracil or FOLFIRINOX (oxaliplatin, irinotecan with leucovorin and infusion of short duration of 5-FU). Unfortunately, the disease frequently progresses as a result of resistance; sometimes the disease is already in an unresectable or locally advanced state at the time of diagnosis.

Next generation sequencing studies have allowed the identification of new mutations and the classification of PDA in subgroups with prognostic and predictive value, in addition to the identification of new therapeutic targets. For example, the combined FOLFIRINOX regimen increases survival in patients with molecular alterations in *BRAC1*, *BRAC2* or *PALB2*, present in the unstable subtype of the Waddell et al. classification [12]. These tumors have a higher sensitivity to platinum and PARP inhibitors, such as olaparib [87]. Mutations in *PI3KCA* and *EGFR* allowed the use of specific inhibitors, such as erlotinib (EGFR inhibitor), but with limited benefits individually or in combination with gemcitabine [88]. It is therefore essential to identify new targets with therapeutic potential, and epigenetic regulators emerge as promising targets. The JQ1 inhibitor, specific for the chromatin remodeler BRD4, belonging to the class of bromodomain and extra terminal (BET) domain inhibitors (BET domains are responsible for recognizing histone acetylation) has shown a therapeutic potential in orthotopic implant models [89] and in GEMM in combination with gemcitabine and histone deacetylase inhibitors such as vorinostat [90]. Among the chromatin remodelers, BPTF stands out, a member of the NURF complex and necessary for the transcriptional activity of the c-MYC oncogene. Its inhibition has proven its therapeutic potential in a PDA mouse model driven by c-MYC (*Ela1-c-MYC*), reducing cell proliferation and tumor volume [91].

Immunotherapy treatments have not demonstrated relevant activity, although the classification proposed by Bailey et al. described a specific immunogenic subtype. The low response may be due to the high desmoplastic reaction, poor tumor vascularization and the hypoxic environment present in the tumor microenvironment together with different genetic alterations, for example in KRAS and *c-MYC*, which prevent the activation of pathways related to IFNα and PD-L1, favoring immune suppression [92]. Inhibition of CXCR4 in combination with pembrolizumab (anti-PD-1) has shown promising results in clinical trials [93]. We require strong predictive markers for these treatments. In this direction, a recent study by Wartenberg et al. that combined NGS analysis with immunohistochemistry proposes three subtypes with different immune characteristics: (i) immune-escaped with few tumor lymphocytes in the stroma (low expression of CD3, CD4, CD8 and increased T regulators with FOXP3), mutations in *KRAS* and worse prognosis; (ii) immune-rich with abundant CD4, CD8 and CD3 lymphocytes, and B cells but proportionally few FOXP3 lymphocytes. Molecularly, this subtype presents mutations in *KRAS*, and less frequent mutations in *CDKN2A*, *SMAD4* and *PIK3CA* than the immuno-escaped and is associated with better survival; (iii) immuno-exhausted, with high percentage of lymphocytes (CD3, CD4 and CD8) and CD8/FOXP3 ratio. The expression of PD-L1 is high and associated with mutations in *PIK3CA* and *JAK*. Its prognosis is similar to immune-escaped. These new studies may be used to better identify groups of patients susceptible to a better response. Meanwhile, other alternatives are the combination of immunotherapy with stromal modulation, Feig et al. combined the use of anti-PD-L1 with FAP inhibitors to facilitate the recruitment of effector T lymphocytes [94], and Jiang et al. demonstrated how focal adhesion kinase (FAK) inhibition favors the action of anti-PD-L1 therapy, increases the levels of CD8 cells and improves survival in mouse models [95]. Finally, Olive et al. and Nagathihalli et al. demonstrated how the specific inhibition of the Hedgehog (Hh) or STAT3 pathway, respectively, can reduce the desmoplastic reaction and enhance tumor vascularization and therefore the response to gemcitabine [96,97].

## 8. Future Perspective

In this review, we focus on recently published data that describe the molecular bases of this pathology with clinical implications. The new findings come from next generation sequencing genomic analyses, and computational tools that have allowed the distinction of different molecular subtypes. The heterogeneity of this tumor is reflected in the response to treatments—some patients show a limited response followed by progression, others show a stable response and later relapse and others do not respond at all—and in the described subtypes. It is crucial to combine both factors to improve patient survival. In this regard, two phase II clinical trials are currently underway. The aim of clinical trial NCT04683315 is to discriminate the efficacy of regimens of FOLFIRINOX in patients with “classical subtype” versus gemcitabine/nab-paclitaxel in patients with “basal subtype”, and the clinical trial NCT03977233 is evaluating the use of FOLFORINOX in neoadjuvant chemotherapy and assessing the efficacy depending on tumor and stromal molecular subtype. We believe that the design of clinical trials that establish targeted therapies based on morphological and molecular characteristics present in the PDA subtypes will have a major impact on patient survival.

## Figures and Tables

**Table 1 ijms-22-02077-t001:** Histological classification of pancreas neoplasms.

Benign	Precursors	Malignant
Serous cystadenoma	Pancreatic intraepithelial neoplasia: PanIN 1, 2 or 3	Serous adenocarcinoma Pancreatic ductal adenocarcinoma (PDA)
	Mucinous cystic neoplasia (MCN) with low- or high-grade dysplasia (Mucinous cystadenoma)	Mucinous cystic neoplasia (MCN) with invasive carcinoma (PDA)
	Intraductal papillary mucinous neoplasm (IPMN)	Intraductal papillary mucinous neoplasm with invasive carcinoma (PDA, colloid carcinoma, etc.)
	Pancreatic intraductal oncocytic papillary neoplas (IOPN)	IOPN with associated invasive carcinoma
	Intraductal tubular papillary neoplasia (ITPN)	ITPN with invasive carcinoma
		Acinar cell carcinoma (ACC)
		Pancreatoblastoma
Pseudopapillary solid neoplasm		Solid pseudopapillary neoplasm with recurrence
Mature teratoma		Immature teratoma

**Table 2 ijms-22-02077-t002:** PDA classification based on NGS studies.

Collisson et al.	Moffit et al.	Bailey et al.	Puleo et al.	Chan-Sang-Yue et al.	Waddell et al.
Classic Mesenchymal Exocrine	Classic Basal	Progenitor Squamous ADEX	Pure classical Pure basal-like	Classic A and B Basal-like A and B	Stable with local rearrangements Scattered rearrangements Unstable
		Immunogenic	Immune Classical		
	Normal/activated stroma		Stroma activated Desmoplastic		

**Table 3 ijms-22-02077-t003:** Genetic syndromes associated with an increased risk of pancreatic cancer with family grouping.

Syndrome	Inheritance Mode	Gene	PDA Risk (Mean Age)
Lynch syndrome	Autosomal dominant	*MSH2* (2p), *MLH1* (3p)	1.3–4% (70 years)
Familial breast cancer (*BRCA2*) and Fanconi anemia (*FANCC* and *FANCG*)	Autosomal dominant	*BRACA2* (13q); *PALB2* (16p); *FANCG* (9p); *BRCA1* (17q)	3.5–10%
X Family	Autosomal domina	*PALLADIN* (4q)	Unknown incidence
Familial melanoma syndrome	Autosomal dominant	*CDKN2A* (9p)	10–17%
Hereditary pancreatitis	Autosomal dominant or recessive	*PRSS1* (7q) *SPINK1* (5q)	25–40% (60 years)
Peutz–Jeghers	Autosomal dominant	*STK11*	30–60% (70 years)
Familial pancreatic cancer	Autosomal dominant	Unknown SNP alterations postulated (telomerase, *NR5A2*, 13q22.1, 15q14) Locus 9q34 of the blood group	9–38% (80 years) Group AB0 (group 0 phenotype has less risk than blood groups A and B)

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
