# Peer review of "Molecular Alterations in Pancreatic Cancer: Transfer to the Clinic"

_ijms, 2021, doi:10.3390/ijms22042077_

Round 1
Reviewer 1 Report
The paper "Molecular Alterations In Pancreatic Cancer: Transfer To The Clinic" tried to explore different molecular alterations both in the precursors and in the invasive pancreatic cancer and their implications and applications in clinical management of the patients with this type of cancer. Authors performed a comprehensive literature's review on the current knowledge and the possible future perspectives in this topic. I think that the manuscript is well-written and conclusions substained by the study.
Authors performed a review of molecular alterations in pancreatic cancer and their clinical implications both for the prognosis and treatment strategy.
The topic is interesting and the paper is interesting for the readers of IJMS.
Author Response
We really appreciate the positive comments of the reviewer and we have carefully checked the English style
Reviewer 2 Report
Reviewer’s report:
In this manuscript, the authors summarize the current state of knowledge on pancreatic ductal adenocarcinoma molecular alterations and classifications and comment on the possibilities of its translation into the clinical setting. The report is well–written, clear and the chapters are well-defined. I recommend it for publication after minor revisions :
Line 28 : « Pancreatic ductal adenocarcinoma (PDA) is a devastating disease with a survival rate of less than 5%. » I suggest that authors re-check the percentage and add a recent reference (Siegel et al, 2021).
Line 36 : endocrine pancreas tumours- change to endocrine pancreatic tumors.
Line 38 :
« Pancreatic cancer development shows a strong association with the consumption of tobacco in cigarettes that increase the risk of suffering the disease in two or three times being proportional to the number of packs consumed annually. » Sentence is not very clear, please re-phrase.
Line 301 : Please correct the name of the author to « Waddell et al » (check reference 11)
Line 314 :
« Endocrine/exocrine characterized by the over-expression of genes related pancreatic linage differentiation. A huge handicap of these studies, except the carry out by Moffitt et al, was the low tumour cellularity due to the high degree of stromal cell infiltration. «
Please correct « linage » to « lineage ».
Additionally, I think it would be of added value for this review mentioning in this paragraph and in table 2 the study of Puleo et al. (Gastroenterology, 2018). This study overcomes the limitations authors mentioned a line 316, as it was performed on a large, multicenter cohort of resected pancreatic ductal adenocarcinoma using prospectively and consecutively collected FFPE samples (FFPE being more amenable for use in clinical diagnostic). By deconvoluting normal, tumoral, and microenvironment transcriptomic signals composing PDA tissue, they identified 5 PDA subtypes and characterized immune and stromal patterns for prognostication and patient stratification for future clinical trials. Their work additionally suggests that ADEX subtype is possible contamination by normal exocrine pancreas rather than an actual subtype.
Author Response
Reviewer 2
Reviewer’s report:
In this manuscript, the authors summarize the current state of knowledge on pancreatic ductal adenocarcinoma molecular alterations and classifications and comment on the possibilities of its translation into the clinical setting. The report is well–written, clear and the chapters are well-defined. I recommend it for publication after minor revisions:
We appreciate the comment of the reviewer and the paragraph “Line 28: « Pancreatic ductal adenocarcinoma (PDA) is a devastating disease with a survival rate of less than 5%. »” have been change to “Pancreatic ductal adenocarcinoma (PDA) is a devastating disease with a survival rate of less than 10%” The reference (Siegel et al, 2021) has been included.
The error in Line 36: endocrine pancreas tumours has been changed to “endocrine pancreatic tumours”
We agree completely with referee regarding the paragraph in Line 38: « Pancreatic cancer development shows a strong association with the consumption of tobacco in cigarettes that increase the risk of suffering the disease in two or three times being proportional to the number of packs consumed annually. » It has been re-phrase to: Line 38: Pancreatic cancer development shows a strong association with the consumption of tobacco in cigarettes that increase the risk of suffering the disease in two or three times. This risk is proportional to the number of packs consumed annually.
The error in Line 301 has been corrected to Waddell et al
In the following paragraph Line 314: “Endocrine/exocrine characterized by the over-expression of genes related pancreatic linage differentiation. A huge handicap of these studies, except the carry out by Moffitt et al, was the low tumour cellularity due to the high degree of stromal cell infiltration”.
We have change « linage » to « lineage ».
We really thank the reviewer for the suggestion, and we have included the study of Puleo et al. (Gastroenterology, 2018) adding the following paragraph: Line 322: These results were validated by Puleo et al in a series of 309 PDA that were analysed by DNA sequencing and transcriptomic profile. By deconvoluting normal, tumoral, and microenvironment transcriptomic signals present in the tumour microenvironment, the authors identified five different subtypes with prognostic value: i) Pure basal like; ii) Stroma activated; iii) Desmoplastic; iv) Pure classical and v) Immune Classical. This work also supports the idea that the ADEX subtype is a possible contamination by normal exocrine pancreas rather than a real subtype.

Reviewer 3 Report
The authors present a comprehensive review on molecular alterations in the pancreas, which ultimately lead to the development of pancreatic cancer. I think the review is well-written and a valuable addition to the field. I have some minor suggestions to improve the paper, which can be found below. - The authors could consider adding a short paragraph on the general role of molecular alterations in the development of cancer, i.e. describing the role of tumor-suppresor and proto-oncogenes. - In addition, authors could summarize currently ongoing trials in the field of precision-treatment of pancreatic cancer and add these to the 'Future Perspectives' paragraph. - Authors are encouraged to carefully review the paper for small inconsistencies such as family pancreatic cancer (familial is the right spelling), double-spaces (line 260). - If possible, please outline tables on a single page. Authors could consider changing the page setting to landscape.Author Response
Reviewer 3
The authors present a comprehensive review on molecular alterations in the pancreas, which ultimately lead to the development of pancreatic cancer. I think the review is well-written and a valuable addition to the field. I have some minor suggestions to improve the paper, which can be found below.
We really appreciate the comments of the reviewer and we have incorporated all the suggestions.
- The authors could consider adding a short paragraph on the general role of molecular alterations in the development of cancer, i.e., describing the role of tumor-suppresor and proto-oncogenes.
Line 82: We have added the following paragraph: Tumour progression is the consequence of the accumulation of multiple genetic alterations that disrupt normal biological pathways. Oncogenic mutations, due to point mutations, amplifications or translocations are responsible of the different tumour hallmarks, such as sustained proliferation, metabolic reprogramming, angiogenesis, inflammation and metastasis. Loss of function in tumour suppressor genes due to mutations, deletions or promoter inactivation contributes to tumour progression through the alteration of pathways involved in DNA repair and apoptosis that lead ultimately to genome instability.
Oncogenes and tumour suppressors are equally altered in PDA,
- In addition, authors could summarize currently ongoing trials in the field of precision-treatment of pancreatic cancer and add these to the 'Future Perspectives' paragraph.
We agree about the importance of point the current ongoing trials and we have included the following paragraph: Line 486: In this regard, two Phase II clinical trials are nowadays on-going. The clinical trial NCT04683315 tries to discriminate the efficacy of regimens of FOLFIRINOX in patients with “classical subtype” versus Gemcitabine/nab-paclitaxel in patients with “basal subype” and the clinical trial NCT03977233 evaluates the use of FOLFORINOX in neoadjuvant chemotherapy and asses the efficacy depending on tumour and stromal molecular subtype. We believe that the design of clinical trials that establish targeted therapies based on morphological and molecular characteristics present in the PDA subtypes, will have a major impact in patient survival.
- Authors are encouraged to carefully review the paper for small inconsistencies such as family pancreatic cancer (familial is the right spelling), double-spaces (line 260).
We thank the appreciation we have changed all the inconsistencies.
- If possible, please outline tables on a single page. Authors could consider changing the page setting to landscape.
All tables are now in landscape.
